# The Safety and Efficacy of Combining Saxagliptin and Pioglitazone Therapy in Streptozocin-Induced Diabetic Rats

**DOI:** 10.3390/biomedicines11123300

**Published:** 2023-12-13

**Authors:** Ahmed Mohamed Othman, Ibrahim Ashour Ibrahim, Samy M. Saleh, Dina M. Abo-Elmatty, Noha M. Mesbah, Asmaa R. Abdel-Hamed

**Affiliations:** 1Department of Biochemistry, Faculty of Pharmacy, Suez Canal University, Ismailia 41522, Egyptnoha_mesbah@pharm.suez.edu.eg (N.M.M.); 2Department of Biochemistry, Faculty of Veterinary Medicine, Suez Canal University, Ismailia 41522, Egypt

**Keywords:** diabetes mellitus, Saxagliptin, Pioglitazone, miRNA-29a, oxidative stress

## Abstract

Background: Type 2 diabetes mellitus (T2DM) is a chronic progressive disease due to insulin resistance. Oxidative stress complicates the etiology of T2DM. Saxagliptin is a selective dipeptidyl peptidase-4 (DPP-4) inhibitor, while Pioglitazone is a thiazolidinedione insulin sensitizer. This study aimed to assess the effect of Saxagliptin and Pioglitazone monotherapy and combination therapy on the biochemical and biological parameters in streptozotocin (STZ)-induced diabetic rats. Methods: The study included thirty-five male albino rats. Diabetes mellitus was induced by intraperitoneal STZ injection (35 mg/kg). For a 1-month duration, rats were divided into five groups. Glucose homeostasis traits, lipid profiles, kidney functions, liver enzymes, and oxidative stress markers were measured. Gene expression of miRNA-29a, phosphoenolpyruvate carboxykinase (PEPCK), phosphoinositide-3-kinase (PI3K), and interleukin 1 beta (IL-1β) was assessed using qRT-PCR. Results: At a 1-month treatment duration, combination therapy improves oxidative stress markers more than either drug alone. The combination therapy had significantly higher levels of SOD, catalase, and GSH and lower levels of MDA compared to the monotherapy. Additionally, the diabetic group showed a significant increase in the expression levels of miRNA-29a, PEPCK, and IL-1β and a significant decrease in PI3K compared to the normal control group. However, combination therapy of Saxagliptin and Pioglitazone was more effective than either Saxagliptin or Pioglitazone alone in reversing these results, especially for PEPCK and IL-1β. Conclusions: Our findings revealed that combining Saxagliptin and Pioglitazone improves glycemic control and genetic and epigenetic expression profiles, which play an essential regulatory role in normal metabolism.

## 1. Introduction

Diabetes mellitus is a metabolic disorder characterized by chronic hyperglycemia [1]. Diabetes mellitus is associated with disturbances in the metabolism of carbohydrates, protein, and lipids [2]. Type 2 diabetes mellitus (T2DM) is a chronic progressive disease. It is mainly due to the gradual loss of functional beta cells and insulin resistance [3]. T2DM is a multifactorial disease due to environmental and genetic factors [4].

The etiology of T2DM is complicated by oxidative stress, leading to an increased risk of metabolic syndrome, inflammation, endothelial dysfunction, and insulin resistance [5]. Insulin resistance is characterized by dyslipidemia, which includes hypertriglyceridemia, elevated levels of low-density lipoprotein cholesterol (LDL-C), elevated free fatty acids, and decreased levels of high-density lipoprotein cholesterol (HDL-C) [6]. 

MicroRNAs (miRNAs) are a class of small endogenous single-stranded, non-coding RNAs that regulate about one-third of the genes in the human genome post-transcriptionally [7]. miRNAs play a critical role in restoring normoglycemia in the human body by regulating the development and differentiation of beta-cell and insulin secretion and actions on the target tissues [8]. miRNAs target genes implicated with insulin exocytosis and beta-cell survival. Therefore, insulin resistance is the central mechanism of miRNA-mediated DM [9]. A meta-analysis conducted on T2DM patients and rodent models showed that miRNA-29a expression was the highest in insulin-sensitive tissues [10]. Another study showed that the expression of miRNA-29a was significantly increased in the liver, muscle, and adipose tissues of diabetic rodents [11]. Subsequently, there was a relation between miRNA-29a overexpression and the incidence of insulin resistance in T2DM [12].

Phosphoenolpyruvate carboxykinase (PEPCK) catalyzes the first committed phase in hepatic gluconeogenesis [13]. Glucagon, catecholamines, and glucocorticoids normally induce PEPCK expression during fasting and in response to stress, while its expression is primarily inhibited by glucose-induced insulin secretion during feeding [14]. Inhibition of insulin secretion, either due to intermittent insulin injection in type I diabetics or insulin resistance in type II diabetics, leads to hyperglycemia and chronic complications [15]. As a result, defining a molecular mechanism for insulin inhibition of PEPCK gene transcription has been a significant research goal, as it would enable the production of drugs to prevent diabetics from experiencing episodic rises in circulating glucose [15]. PEPCK is considered a sensitive marker to check the effects of compounds on the regulation of glucose levels [16].

Phosphoinositide-3-kinase (PI3K)/AKT signaling plays a critical role in cellular physiology [17]. It mediates growth factor signals and essential cellular processes such as glucose homeostasis, lipid metabolism, protein synthesis, and cell proliferation and survival [17].

Interleukin-1 (IL-1) is generated by cells in response to glucose stimulation. Furthermore, IL-1 induces the synthesis of its cytokines in cells and attracts macrophages, which may serve as a secondary source of IL-1 and other cytokines [18]. Although it is currently uncertain if inflammatory responses are a primary cause or a secondary effect in T2DM development, therapies targeting IL-1 have shown promising results in various clinical trials [19].

Saxagliptin is a selective dipeptidyl peptidase-4 (DPP-4) inhibitor [20]. It increases glucagon-like peptide-1 levels, motivates insulin production, and lowers glucose levels by activating the incretin hormones, glucagon-like peptide-1, and glucose-dependent insulinotropic polypeptide [21]. Pioglitazone, an insulin sensitizer, is one of the thiazolidinediones (TZDs), and it is a peroxisome proliferator-activated receptor-gamma (PPAR-γ) agonist [22]. Pioglitazone enhances insulin sensitivity as it regulates the expression of many genes concerned with lipid and carbohydrate metabolism [23]. Pioglitazone decreases PEPCK expression and the production of several mediators from adipocytes that may cause insulin resistance [24]. Pioglitazone is reported to downregulate the hepatic miRNA-29a expression level, reduce insulin resistance, and restore normoglycemia in animal models with insulin resistance and diabetes [25]. On the other hand, Linagliptin, a DPP-4 inhibitor, did not show any alterations in the levels of miRNA-29a expression when compared to angiotensin-receptor blocker telmisartan or a placebo [26].

Previous literature showed that adding DPP-4 inhibitors to Pioglitazone was well tolerated, did not cause hypoglycemia, and did not significantly worsen Pioglitazone-induced weight gain [27]. In people with T2DM, combining Sitagliptin or Vildagliptin with Pioglitazone can be an effective treatment option [27] and improves glycemic control [28]. In a 76-week trial on patients with type 2 diabetes mellitus (T2DM), there was no rise in the cardiac or cardiovascular safety signals in the groups receiving Saxagliptin and thiazolidinedione medication [29]. Also, there is no Drug–Drug Interaction between Pioglitazone and Saxagliptin. They do not require changing the Saxagliptin dosage to be taken simultaneously [30]. This study aimed to evaluate the benefit of Saxagliptin and Pioglitazone combination therapy on biochemical and biological parameters in streptozotocin (STZ)-induced diabetic rats.

## 2. Materials and Methods

### 2.1. Experimental Animals

Thirty-five male albino rats weighing 90–110 g were purchased from the National Research Institute (Cairo, Egypt). Animals were housed in an animal room in separate metal cages with temperature control (20–24 °C) and 12 h light–dark cycles. Rats received standard rat chow and tap water. An adaptation period of 1 week was allowed before beginning the experiment. The Guide for the Care and Use of Laboratory Animals was followed for keeping and using experimental animals (National Research Council, 2011). The Ethics Committee at the Faculty of Pharmacy, Suez Canal University (Ismailia, Egypt) approved all experimental protocols (ethics code #202105RA1).

### 2.2. Experimental Design

The rats were allocated into two dietary regimens; 7 rats (group A) were fed a standard balanced diet (12% calories as fat), and 28 rats (groups B, C, D, and E) were fed a high-fat diet (HFD; 58% calories as fat) [31]. After 4 weeks of dietary manipulation, 28 rats fed HFD were injected intraperitoneally (i.p) with STZ (35 mg/kg) [32] dissolved in a 0.1 M sodium citrate buffer (pH 4.5) as a single dose. After 10 days from STZ administration, blood samples were obtained from the tail tip, and then the glucose level was measured using the glucometer method (Gluco Dr-All Medicus Co., Ltd., Anyang-si, Republic of Korea). Only rats with fasting blood glucose ≥ 200 mg/dL were enrolled [33]. The studied groups were divided into five groups of 7 rats each as follows: group A, normal control; group B, STZ-diabetic; group C, STZ-diabetic treated with 20 mg/kg of Pioglitazone; group D, STZ-diabetic treated with 10 mg/kg of Saxagliptin; and group E, STZ-diabetic treated with a combination of 10 mg/kg of Saxagliptin and 20 mg/kg of Pioglitazone (Figure 1). Pioglitazone and Saxagliptin were dissolved in distilled water for oral gavage administration to diabetic rats once daily for 1 month starting from day 11 after induction of diabetes [24].

### 2.3. Drugs

Streptozotocin (S0130-1G) was purchased from Sigma Aldrich, Cairo, Egypt. Diabtein (Pioglitazone) (23390/04) was purchased from Unipharma, Cairo, Egypt, and Onglyza (Saxagliptin) (PLGB 17901/0337) was purchased from AstraZeneca, Cambridge, UK. 

### 2.4. Blood Sampling and Biochemical Analysis

At the end of the experiment, rats were anesthetized with thiopental sodium (50 mg/kg). Retro orbital blood samples were withdrawn. Rats were sacrificed, and liver tissues were collected and kept at −80 °C for RNA extraction. Blood samples were centrifuged at 3000 rpm for 15 min to separate serum and stored at −80 °C for assaying biochemical parameters. 

### 2.5. Glucose Hemostasis Traits and Lipid Profiles

The serum glucose level was measured using the enzymatic colorimetric method (Biodiagnostic kit, Giza, Egypt, Catalog No. GLU109240), performed in two replicates. The following equation obtained glucose concentration: Glucose concentration = (A_sample_/A_standard_) × Standard Conc. Serum insulin was determined using a rat insulin (INS) ELISA kit (Wuhan Fine Biotech Co., Ltd., Wuhan, China, Catalog No. ER1113) and was used in two replicates (standard curve range: 78.125–5000 pg/mL, sensitivity: <46.875 pg/mL, intra-assay: CV < 8%, inter-assay: CV < 10%) as per manufacturer procedures. Two indirect indices were calculated. First, the Homeostasis Model Assessment-Insulin Resistance (HOMA-IR) was calculated using the equation (glucose × insulin)/405 [34]. Second, the quantitative insulin sensitivity check index (QUICKI) was calculated using the equation 1/(log insulin + log glucose) [35].

Lipid profiles, including serum triglycerides (TGs), total cholesterol (TC), and high-density lipoprotein cholesterol (HDL-C), were measured in two replicates with enzymatic colorimetric methods using Biodiagnostic kits, Egypt (Catalog No. TG117249, Catalog No.CH0104200, and Catalog No. MG266001, respectively). Low-density lipoprotein cholesterol (LDL-C) was calculated according to Friedewald [36].

### 2.6. Liver and Kidney Functions

Liver enzymes, including alanine aminotransferase (ALT) and aspartate aminotransferase (AST), were measured with enzymatic colorimetric methods using Biodiagnostic kits, Egypt (Catalog No. AL 10 31 (45) and Catalog No. AS 10 61 (45), respectively).

Kidney function tests, including creatinine and urea, were measured with enzymatic colorimetric methods using Biodiagnostic kits, Egypt (Catalog No. CR 12 50 and Catalog No. UR 21 10, respectively).

### 2.7. Oxidative Stress Parameters

According to the manufacturer’s instructions, all the oxidative stress markers were assessed in the liver tissue homogenate of the studied groups with enzymatic colorimetric methods. Superoxide dismutase (SOD) was measured using a Biodiagnostic kit, Egypt (Catalog No. SD2521). Catalase was assessed using a Biodiagnostic kit, Egypt (Catalog No. CA2517). Malondialdehyde (MDA) was determined using a Biodiagnostic kit, Egypt (Catalog No. MD 25 29). Glutathione reduced (GSH) was measured using a Biodiagnostic kit, Egypt (Catalog No. GR2511). 

### 2.8. RNA Extraction and Assaying Gene Expression

According to the manufacturer’s protocol, total RNA, including miRNAs, was extracted from liver tissue homogenates using a miRNeasy Mini kit (Qiagen, Hilden, Germany, Catalog No. 217004). RNA concentration was determined spectrophotometrically using a NanoDrop 1000 spectrophotometer (NanoDrop Tech, Wilmington, DE, USA). 

### 2.9. MicroRNA-29a Reverse Transcription and qPCR Detection

MicroRNA-29a expression was quantified with qRT-PCR. Reverse transcription (RT) was performed using a TaqMan^®^ microRNA reverse transcription kit (Applied Biosystems, Waltham, MA, USA, Catalog No. 4366596) and specific miRNA primers miRNA-29a 5× and 18S rRNA 5× (Applied Biosystems, USA, Catalog No. 4427975) according to the manufacturer’s instructions. 18S rRNA was used as an endogenous control. The RT reaction was performed in a Mastercycler Gradient thermocycler (Eppendorf, Hamburg, Germany) at 16 °C for 30 min, 42 °C for 30 min, 85 °C for 5 min, then held at 4 °C.

Real-time PCR was carried out in an AB 7500HT instrument with SDS Software version 2.1.1, using TaqMan^®^ Universal Master Mix II, with UNG (Applied Biosystems, USA, Catalog No. 4440038), and TaqMan^®^ assays miRNA-29a 20× and 18S rRNA 20× (Applied Biosystems, USA, Catalog No. 4427975) according to the manufacturer’s instructions. All reactions were run in duplicate using the following cycling conditions: 95 °C for 10 min, followed by 40 cycles of 95 °C for 15 s and 60 °C for 1 min. 

Expression of miRNA-29a was reported as a ΔCt value. The ΔCt value was calculated by subtracting Ct values of the (18S rRNA) endogenous control from the Ct values of the target miRNA (miRNA-29a). The relative expression of miRNA-29a was analyzed using the standard 2^−ΔΔCt^ method [37].

### 2.10. PI3K, PEPCK, and IL-1β mRNA Reverse Transcription and qPCR Detection

According to the manufacturer’s instructions, RNA was reversely transcribed to cDNA using the TaqMan^®^ high-capacity cDNA reverse transcription kit (Applied Biosystems, USA, Catalog No. 4368813). RT reactions were carried out in a Mastercycler Gradient thermocycler (Eppendorf, Hamburg, Germany) at 25 °C for 10 min, 37 °C for 120 min, 85 °C for 5 min, then held at 4 °C.

mRNA relative expression of PI3K, PEPCK, IL-1β, and 18S rRNA was quantified using the primer assays illustrated in Table 1 and TaqMan 2× universal PCR master mix with UNG (Applied Biosystems, USA, Catalog No. 4440034) according to the manufacturer’s instructions. Gene expression was normalized to the 18S rRNA gene. Real-time PCR was carried out in an AB 7500HT instrument with the SDS Software version 2.1.1 system (Applied Biosystems, USA) as follows: 50 °C for 2 min, 95 °C for 10 min for polymerase activation followed by 40 cycles of 95 °C for 15 s denaturation, 60 °C for 1 min annealing, and 60 °C for 1 min extension. ΔΔCt and fold change were calculated to determine the relative gene expression of PI3K, PEPCK, and IL-1β mRNAs [37].

### 2.11. Statistical Analysis

The Statistical Package for Social Sciences (SPSS Inc., Chicago, IL, USA) version 22 software was used to manage the data. The data were checked for normality using the Kolmogorov–Smirnov test. The results were presented as the mean ± standard error (SE). A one-way analysis of variance, ANOVA, followed by Tukey’s multiple comparison test, was employed for the statistical analysis. A correlation analysis using Pearson rank correlation was performed. A value of *p* < 0.05 was considered to be statistically significant. 

## 3. Results

### 3.1. Biochemical Analysis

The biochemical analysis revealed that, compared to the normal control group, the STZ-induced diabetic group showed a significant increase in FBG (343.86 vs. 89.71 mg/dL), HOMA-IR (5.3 vs. 2.52), TC (173.34 vs. 92.65 mg/dL), TG (156.8 vs. 66.86 mg/dL), and LDL-C (136.5 vs. 39.51 mg/dL), all at *p* < 0.001. In addition, the STZ-induced diabetic group showed a significant decrease in fasting insulin (6.31 vs. 11.51 µU/mL), QUICKI (0.3 vs. 0.33), and HDL-C (22.38 vs. 43.6 mg/dL) compared to the normal control group at *p* ˂ 0.001 (Table 2). Treatment with a combination of Saxagliptin and Pioglitazone markedly improved these biochemical markers when compared to Pioglitazone or Saxagliptin monotherapy. Compared to the Pioglitazone-treated group, the addition of Saxagliptin to Pioglitazone reduced FBG (155.29 vs. 179.86 mg/dL), HOMA-IR (3.03 vs. 3.17), and LDL-C (47.58 vs. 59.83 mg/dL) and elevated fasting insulin (8.04 vs. 7.06 µU/mL), and HDL-C (38.95 vs. 34.62 mg/dL) (Table 2). 

Regarding liver and kidney functions, there was a significant increase in serum levels of ALT, AST, urea, and creatinine in the STZ-induced diabetic group compared to the normal control group at *p* < 0.001 (Table 3). However, treatment with combination therapy of Saxagliptin and Pioglitazone significantly improved these functions compared to monotherapy groups (Table 3).

### 3.2. Oxidative Stress

As shown in Table 3, the cellular antioxidant and anti-inflammatory defense mechanism showed a significant decrease in levels of SOD (2031 vs. 4680.14 U/g tissue), catalase (55.16 vs. 121.03 U/g tissue), and GSH (14.02 vs. 37.0 mmol/g tissue), and in the liver homogenate, there was a significant increase in MDA (429.89 vs. 259.34 nmol/g tissue) in the STZ-induced diabetic group compared to the normal control group at *p* < 0.001. However, treatment with combination therapy of Saxagliptin and Pioglitazone significantly improves the oxidative stress parameters better than Saxagliptin or Pioglitazone monotherapy. The group treated with Saxagliptin and Pioglitazone showed a significant increase in levels of SOD, catalase, and GSH and a significant decrease in MDA at *p* < 0.001 compared to the groups treated with either Pioglitazone or Saxagliptin (Table 3).

### 3.3. Genetic and Epigenetic Parameters

At a 1-month treatment duration, in the diabetic liver tissues, there was a significant increase in the expression of miRNA-29a, PEPCK, and IL-1β genes at *p* < 0.001 and a decrease in expression of the P13K gene when compared to the normal control group at *p* = 0.02 (Figure 2). The combination therapy of Saxagliptin and Pioglitazone achieved some improvements in genetic and epigenetic parameters compared to treatment with Saxagliptin or Pioglitazone alone. The treated groups exhibited a significant upregulation of PIK3R1 expression and downregulation of PCK1, IL1B, and MIR 29a compared to the diabetic control rats (Figure 2).

The correlation between genetic and epigenetic parameters is shown in Table 4. There was a strong positive correlation between miRNA-29a, PEPCK, and IL-1β. Also, there was a strong negative correlation between these previous genes and PI3K based on the change in the expression levels according to the fold change values (Table 4). 

## 4. Discussion

T2DM requires effective therapeutic targets [38]. The proper management of T2DM may be well achieved using combination therapy rather than monotherapy [39]. 

This study’s biochemical analysis findings are consistent with previous studies into the potential benefits of DPP-4 inhibitors and TZD in treating T2DM. Regarding glycemic control, two previous 24-week studies on human subjects documented that a combination of Saxagliptin and TZD was superior to TZD monotherapy [40]. A recent systematic review and meta-analysis of seven randomized controlled trials found that combining a DPP-4 inhibitor with Pioglitazone was superior to Pioglitazone monotherapy [41]. This is similar to our findings, which revealed a marked reduction in FBG in the group receiving combination therapy compared to monotherapy groups. Moreover, the fasting insulin level and the two indirect indices (HOMA-IR and QUICKI) were notably improved in the group receiving combination therapy than the other groups. This implies that using combination therapy in T2DM improves glycemic control [42]. Concerning lipid profiles, the group receiving combination therapy had a significantly better lipogram than those receiving monotherapy. Pioglitazone regulates lipogenesis, adipokine discharge, and hormones involved in carbohydrate and lipid metabolism [43]. DPP-4 inhibitors decrease hepatic lipid synthesis, sterol regulatory element-binding proteins-2 (SREBP-2) expression, and SREBP-2 activity [44]. So, both Pioglitazone and Saxagliptin tend to restore lipid profiles in T2DM patients. Moreover, Saxagliptin and Pioglitazone appear to have a synergistic effect on lipid profile improvement, as the best results were observed in the group that received combination therapy. In line with our findings, Saxagliptin was reported to improve dyslipidemia [45]. Pioglitazone has also been shown to improve lipid profiles [46].

Regarding oxidative stress, the effect of DPP-4 inhibitors and TZDs has been previously studied individually. One animal study reported similar results to our experiment, where three DPP-4 inhibitors, namely Saxagliptin, Sitagliptin, and Vildagliptin, were found to significantly reduce oxidative stress in STZ-induced diabetic rats [45]. Meanwhile, in a 16-week clinical trial involving 40 patients with advanced diabetic nephropathy, Pioglitazone did not affect reducing oxidative stress [47]. In contrast, after 8 weeks of Pioglitazone administration, there was a partial improvement in the oxidative stress markers in the kidney and liver of diabetic rabbits [48]. The conflicting results of the two studies on Pioglitazone can be attributed to the different disease stages at which the drug was administered. It has been proposed that Pioglitazone may be more effective at earlier stages of disease [47]. It is possible that TZDs, like Pioglitazone, act through both an antidiabetic and an antioxidant mechanism [49]. In this study, the combination therapy of Saxagliptin and Pioglitazone significantly reduced oxidative stress, which is similar to the findings of Refaat et al., who reported a possible benefit of combining a DPP-4 inhibitor (Vildagliptin) and Pioglitazone in lowering MDA and increasing GSH compared to either Pioglitazone or Vildagliptin alone [46]. Regarding SOD, combination therapy of Saxagliptin and Pioglitazone was more efficient in re-elevating the SOD level in STZ-induced diabetic rats than groups receiving monotherapy. Moreover, Pioglitazone monotherapy was superior to Saxagliptin. It has also been proposed that TZDs, such as Rosiglitazone and Pioglitazone, can act by activating PPAR-γ, resulting in increased uncoupling protein 2 (UCP2) expression in the inner mitochondrial membrane, causing its depolarization [50,51]. Eventually, this lowers the levels of superoxide anions, which in turn reduces oxidative stress [52].

The expression of miRNA-29a was significantly reduced in all treated groups, with the most notable decrease in the group that received combination therapy. Overexpression of miRNA-29a targets the key signaling molecule PI3K, which reduces the insulin-inhibitory effect on PEPCK expression and contributes to unrestrained gluconeogenesis [11]. The impacts of TZDs and DPP-4 inhibitors on the miRNA-29 expression are opposed. While Pioglitazone has been shown to suppress miRNA-29a expression, DPP-4 inhibitors have been shown to restore miRNA-29a levels in STZ-induced diabetic mice [25]. Surprisingly, miRNA-29a expression was lower in the combination therapy group than in the healthy control group. Although overexpression of miRNA-29a has been linked to increased circulating blood glucose, insulin resistance, and hyperlipidemia in diabetes, suppression of miRNA-29a has been linked to profibrotic gene overexpression [25,53]. This means miRNA-29a expression should be carefully balanced to avoid potential risks [54]. Therefore, the exaggerated reduction in miRNA-29a expression observed in the combination therapy group may predispose to some unwanted adverse effects for the treatment strategy. 

Similar to our findings, TZDs such as Pioglitazone have been shown to improve glycemic control in insulin-resistant patients by inhibiting the expression of PEPCK and glucose 6-phosphatase genes [55]. Furthermore, Evogliptin has also been shown to downregulate PEPCK expression, resulting in better glycemic control [56]. In contrast to previous findings, Teneligliptin failed to show any significant difference in the expression levels of the two gluconeogenic genes, PEPCK and glucose 6-phosphatase, when compared to diabetic controls, indicating the superiority of Saxagliptin over other DPP-4 inhibitors [57]. 

PI3K is an essential protein of the insulin signaling transduction pathway [58]. One of the primary causes of T2DM is a decrease in PI3K signaling downstream of the insulin receptor [59]. Loss of PI3K signaling in the muscle causes decreased glucose uptake, whole-body glucose intolerance, hyperlipidemia, and obesity, whereas loss of PI3K signaling in the liver causes uncontrolled hepatic gluconeogenesis, hyperinsulinemia, and hyperglycemia [59]. Pioglitazone has been reported to increase the levels of the PI3K enzyme [60]. The upregulation of PI3K expression is thought to be significantly influenced by the activation of the PPAR-γ, a major Pioglitazone target [61]. This agrees with the findings of this study. The effects of DPP-4 inhibitors on PI3K expression have been studied, but the results are conflicting. This study showed that Saxagliptin upregulates PI3K expression compared with the diabetic control group. This agrees with the results of Khedr et al. on Sitagliptin [62] and Zaky et al. on Vildagliptin [63]. On the other hand, Teneligliptin was able to significantly reduce the expression of the PI3K gene when administered early in relation to the degree of steatosis, which is one of the histological changes that occurs in the non-alcoholic fatty liver disease associated with T2DM [57].

In several cases, Pioglitazone and Saxagliptin have previously been shown to reduce inflammation and alter inflammatory cytokine (IL-1β and IL-6) levels [64,65]. The exact mechanism by which Pioglitazone reduces the interleukin levels in T2DM mice is unclear. However, it has been suggested that Pioglitazone, as an antidiabetic, reduces glucose-induced IL-1β production, thereby decreasing the apoptosis of islet β-cells [66]. The in vitro findings indicate that TZDs’ anti-inflammatory properties are due, at least in part, to their ability to induce glucocorticoid nuclear translocation independent of PPAR-γ activity [66]. Meanwhile, Saxagliptin decreases IL-1β by repressing the NOD-like receptor 3 (Nlrp3) inflammasome, an interleukin cytokine-activating protein complex [67]. Khedr et al. reported that Sitagliptin downregulates IL-1β expression, which is consistent with our findings [62]. Furthermore, the combination therapy failed to achieve significantly lower IL-1β levels than the two other interventional groups. This might have been due to the different mechanisms of action of the two drugs.

Finally, in vitro studies have shown that Pioglitazone does not affect the levels of the enzyme cytochrome P450 3A4 (CYP3A4), which is responsible for the conversion of Saxagliptin into the active metabolite 5-hydroxy Saxagliptin [30]. As a result, combining the two drugs appeared to be an effective strategy for treating T2DM.

## 5. Conclusions

The results of this study suggest that Pioglitazone–Saxagliptin combination therapy could be used as a potential alternative treatment in T2DM cases that are resistant to monotherapy options. Furthermore, the two drugs were found to be complementary in their actions. There were no significant Drug–Drug Interactions, avoiding major changes in either drug’s dosage. More research is needed to investigate the mechanisms of action of the two drugs, focusing on the potential applicability of other members of the two drug categories.

## Figures and Tables

**Figure 1 biomedicines-11-03300-f001:**
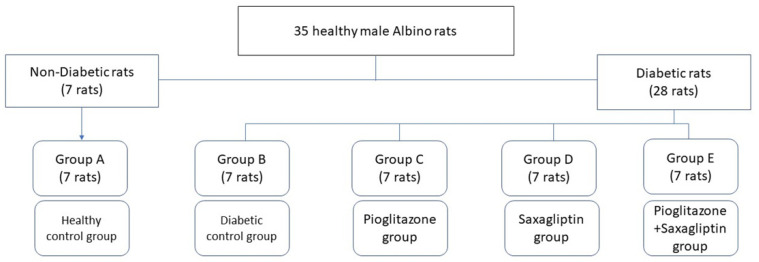
Classification of the studied groups.

**Figure 2 biomedicines-11-03300-f002:**
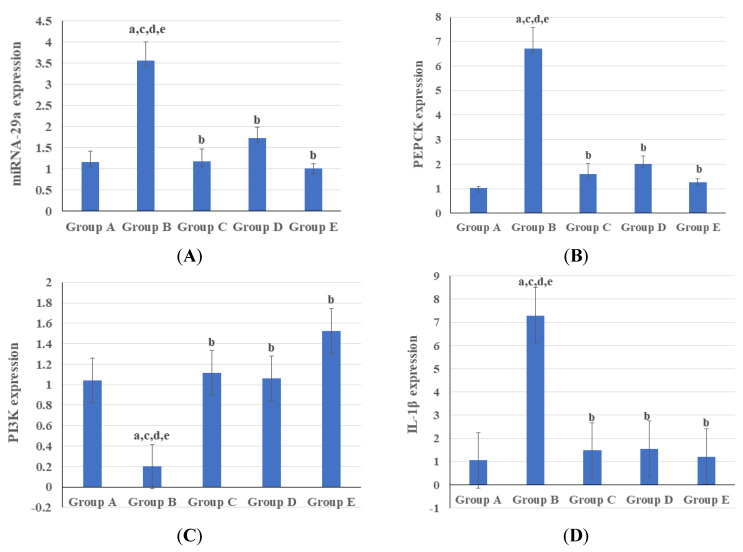
Expression levels of (**A**) miRNA-29a, (**B**) PEPCK, (**C**) PI3K, and (**D**) IL-1β in the studied groups. miRNA-29a, microRNA-29a; PEPCK, phosphoenolpyruvate carboxykinase; PI3K, phosphoinositide-3-kinase; IL-1β, interleukin 1 beta. Group A, normal control; group B, diabetic control; group C, Pioglitazone-treated group; group D, Saxagliptin-treated group; and group E, group received combination therapy of Pioglitazone and Saxagliptin. a–e relates to which groups the group has a significant difference with.

**Table 1 biomedicines-11-03300-t001:** Primers and annealing temperatures used in real-time PCR reactions.

Gene	Catalog No.	Primers	Annealing Temperature, °C
PI3K	4448892	Forward: 5′-GAGCCGAGTTGGAGGAAGCA-3′ Reverse: 5′-CATCCGGGTGTCCATCTGTC-3′	60
PEPCK	4331182	Forward: 5′-CGTTGGGAGCTAGGAGCAAA-3′ Reverse: 5′-CCCATCAGTGTCAGATGCGA-3′	60
Il-1β	4453320	Forward: 5′-GGCGGTTCAAGGCATAACAG-3′ Reverse: 5′-TCAGACAGCACGAGGCATTT-3′	60
18S rRNA	4308329	Forward: 5′-ACGGACCAGAGCGAAAGCAT-3′ Reverse: 5′-TGTCAATCCTGTCCGTGTCC-3′	60

**Table 2 biomedicines-11-03300-t002:** Glucose hemostasis traits and lipid profiles in the studied groups.

Groups	FBG (mg/dL)	Insulin (µU/mL)	HOMA–IR	QUICKI	TC (mg/dL)	TG (mg/dL)	HDL-C (mg/dL)	LDL-C (mg/dL)
Normal control	89.71 ± 4.21	11.51 ± 0.77	2.52 ± 0.11	0.33 ± 0.002	92.65 ± 4.33	66.86 ± 3.2	43.6 ± 1.78	39.51 ± 1.15
Diabetic control	343.86 ± 10.65 ^a, c, d, e^	6.31 ± 0.48 ^a^	5.3 ± 0.27 ^a, c, d, e^	0.3 ± 0.0019 ^a, c, e^	173.34 ± 2.95 ^a, c, d, e^	156.8 ± 6.44 ^a, c, d, e^	22.38 ± 1.49 ^a, c, d, e^	136.5 ± 2.06 ^a, c, d, e^
Pioglitazone group	179.86 ± 5.15 ^a, b, d^	7.06 ± 0.63 ^a^	3.17 ± 0.36 ^b^	0.32 ± 0.005 ^b^	113.63 ± 3.47 ^a, b, d^	84.08 ± 2.62 ^a, b^	34.62 ± 1.39 ^a, b, d^	59.83 ± 1.72 ^a, b, d, e^
Saxagliptin group	210.43 ± 9.02 ^a, b, c, e^	7.82 ± 1.11 ^a^	3.92 ± 0.39 ^a, b^	0.31 ± 0.004 ^a^	129.35 ± 3.24 ^a, b, c, e^	98.41 ± 2.86 ^a, b, e^	28.46 ± 1.1 ^a, b, c, e^	69.2 ± 0.82 ^a, b, c, e^
Saxagliptin + Pioglitazone	155.29 ± 4.25 ^a, b, d^	8.04 ± 0.92 ^a^	3.03 ± 0.27 ^b^	0.32 ± 0.0037 ^b^	107.35 ± 3.59 ^a, b, d^	78.79 ± 2.09 ^b, d^	38.95 ± 1.27 ^b, d^	47.58 ± 1.2 ^a, b, c, d^

Data are presented as mean ± SE. Comparisons were performed with a one-way ANOVA test at *p* < 0.05. *n* = 7 rats per each group. FBG, fasting blood glucose; HOMA-IR, hemostasis model assessment insulin resistance; QUICKI, quantitative insulin sensitivity check index; TC, total cholesterol; TG, triglyceride; HDL-C, high-density lipoprotein cholesterol; LDL-C, low-density lipoprotein cholesterol. a = significantly different from normal control, b = significantly different from diabetic control, c = significantly different from Pioglitazone, d = significantly different from Saxagliptin, e = significantly different from Saxagliptin and Pioglitazone.

**Table 3 biomedicines-11-03300-t003:** Kidney functions, liver enzymes, and oxidative stress markers in the studied groups.

Groups	Urea (mg/dL)	Creatinine (mg/dL)	ALT (U/L)	AST (U/L)	SOD (U/gm Tissue)	Catalase (U/g Tissue)	MDA (nmol/g Tissue)	GSH (mmol/g Tissue)
Normal control	41.16 ± 4.19	0.73 ± 0.04	21.45 ± 1.38	59.39 ± 2.35	4680.14 ± 307.31	121.03 ± 4.74	259.34 ±11.29	37.0 ± 2.31
Diabetic control	109.61 ±4.71 ^a, c, d, e^	1.52 ± 0.11 ^a, c, d, e^	39.35 ± 1.59 ^a, c, d, e^	94.91 ± 3.19 ^a, c, d, e^	2031 ± 193.02 ^a, c, d, e^	55.16 ± 3.02 ^a, c, d, e^	429.89 ±11.37 ^a, c, d, e^	14.02 ± 1.21 ^a, c, d, e^
Pioglitazone group	91.33 ± 3.74 ^a, b, d, e^	0.9 ± 0.08 ^b^	25.19 ± 1.31 ^b^	71.22 ± 3.42 ^a, b, e^	3502.75 ± 63.59 ^a, b^	86.65 ± 2.14 ^a, b, d, e^	339.12 ± 4.93 ^a, b, d, e^	26.25 ± 0.89 ^a, b, d, e^
Saxagliptin group	70.03 ± 4.72 ^a, b, c^	0.81 ± 0.05 ^b^	28.84 ± 0.8 ^a, b, e^	74.59 ± 1.046 ^a, b, e^	2972.92 ± 272.97 ^a, b, e^	72.47 ± 2.45 ^a, b, c, e^	382.86 ± 5.84 ^a, b, c, e^	20.16 ± 1.27 ^a, b, c, e^
Saxagliptin + Pioglitazone	61.26 ± 4.12 ^a, b, c^	0.78 ± 0.07 ^b^	22.09 ± 1.61 ^b, d^	60.64 ±. 0.54 ^b, c, d^	4297.49 ±187.71 ^b, d^	101.4 ± 2.31 ^a, b, c, d^	302.2 ± 3.34 ^a, b, c, d^	33.68 ± 1.06 ^b, c, d^

Data are presented as mean ± SE. Comparisons were performed with a one-way ANOVA test at *p* ˂ 0.05. *n* = 7 rats per each group. ALT, alanine aminotransferase; AST, aspartate aminotransferase; SOD, superoxide dismutase; MDA, malondialdehyde; GSH, reduced glutathione. a = significantly different from normal control, b = significantly different from diabetic control, c = significantly different from Pioglitazone, d = significantly different from Saxagliptin, e = significantly different from Saxagliptin and Pioglitazone.

**Table 4 biomedicines-11-03300-t004:** Correlation between genetic and epigenetic parameters in the studied groups.

	miRNA 29a		PI3K Gene		PEPCK Gene		IL-1β Gene	
	r	*p* Value	r	*p* Value	r	*p* Value	r	*p* Value
miRNA 29a			−0.49	0.003 *	0.74	˂0.001 *	0.79	˂0.001 *
PI3K gene	−0.49	0.003 *			−0.56	0.001 *	−0.54	0.001 *
PEPCK gene	0.74	˂0.001 *	−0.56	0.001 *			0.93	˂0.001 *
IL-1β gene	0.79	˂0.001 *	−0.54	0.001 *	0.93	˂0.001 *		

miRNA, microRNA; PI3K, phosphoinositide-3-kinase; PEPCK, phosphoenolpyruvate carboxy kinase; IL-1β, interleukin beta; r, Pearson rank correlation. * Significantly correlated. Correlation is significant at *p* ˂ 0.05.

## Data Availability

The datasets generated and analyzed during the current study are not publicly available. They can be available after this publication, and they can be available from the corresponding author upon reasonable request.

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
