# Peer review of "The Safety and Efficacy of Combining Saxagliptin and Pioglitazone Therapy in Streptozocin-Induced Diabetic Rats"

_biomedicines, 2023, doi:10.3390/biomedicines11123300_

Round 1

Reviewer 1 Report

Comments and Suggestions for Authors

In this manuscript, Ahmed M. Othman and colleagues evaluate the benefit of combination therapy of Saxagliptin and Pioglitazone on biochemical and biological parameters in streptozotocin-induced diabetic rats. I find the manuscript very interesting and it fits well within the scope of the paper. However, I have a series of major comments to raise with the authors before considering it suitable for publication in Biomedicines.

-          The aim of the work must be clear and incisive, and it must also be well explained throughout the work, especially in the paragraphs of the introduction and discussion;

-          Concerning the streptozotocin-induced diabetic rat model used, the authors should explain in detail why the animals before STZ treatment were fed with an HFD diet for 5 weeks. Why only 5 weeks of an HFD diet?

-          What are the reasons for choosing the STZ dose of 35 mg/kg?

-          Is a single administration of 35 mg/kg necessary to make an animal diabetic?

-          In sections 3.2 and 3.1, the authors refer to the results in Table 3. The work lacks Table 3 as a result;

Comments on the Quality of English Language

Minor editing

Author Response

Dear Reviewer 1,

We would like to thank you for your insightful comments on our manuscript titled "The Safety and Efficacy of Combining Saxagliptin and Pioglitazone Therapy in Streptozocin-Induced Diabetic Rats" (Manuscript ID: biomedicines-2728012).

We appreciate the time and effort you have dedicated to providing us with feedback on our paper, and we are grateful for the insightful comments and helpful suggestions for improving our manuscript. In light of your questions and suggestions, we were able to revise and incorporate important amendments into our manuscript. Please find below our point-by-point response to the comments. We have also uploaded the revised manuscript, with amendments highlighted in yellow. Please note that we sometimes talk about lines that aren't highlighted because they are part of the answer.

We look forward to hearing from you in due time and responding to any further questions and comments you may have.

Sincerely,

Ahmed Othman

This file contains all comments provided by reviewer 1 and the authors' response.

  Comment 1: The aim of the work must be clear and incisive, and it must also be well explained throughout the work, especially in the paragraphs of the introduction and discussion;

Response 1: Thanks a lot , a comprehensive editing done through the manuscript  to clarify the aim of the work

Comment 2: “Concerning the streptozotocin-induced diabetic rat model used, the authors should explain in detail why the animals before STZ treatment were fed with an HFD diet for 5 weeks. Why only 5 weeks of an HFD diet.”

Response 2: Thank you for your insightful comments. The main goal of combining HFD and low dose of STZ is to rapidly simulate the natural history and metabolic progression of the disease from a healthy metabolism to the first stage of "prediabetes and insulin resistance" and then to the second stage of "beta cell failure, frank hyperglycemia, and type 2 diabetes" in a short period of time. In humans, the transition from a healthy metabolic state to a state of prediabetes and insulin resistance is often associated with nutritional overload and obesity. This is why the high fat diet (HFD), western diet, or high energy diet is frequently used to mimic the transition from a healthy metabolism to prediabetes and insulin resistance. At this stage of insulin resistance, the pancreas compensates by producing more insulin (resulting in hyperinsulinemia) to maintain normal glucose levels. In order to progress from prediabetes to the second stage of frank type 2 diabetes and hyperglycemia, a significant portion of pancreatic beta cells must be lost. At this point, STZ appears on the scene to do its job of inducing beta cell death/failure, resulting in relative insulin deficiency [1], [2], [3], [4].

The administration of HFD alone, either for a relatively short time (2-4 weeks) or a relatively

long time (≥ 3 months), to induce diabetes have been heavily investigated in previous studies. It has been shown that HFD alone for short periods can mimic the early stage of the disease and cause hyperinsulinemia, impaired glucose tolerance, a slight increase in non-fasting glucose or even a normal glucose level is maintained, and insulin resistance in peripheral target tissues such as liver and muscle. These models with insulin resistance are very responsive to treatment with drugs that increase peripheral tissues' sensitivity to insulin, such as pioglitazone and metformin [1], [2], [3], [4]. On the other hand, models that rely solely on STZ alone miss the early metabolic changes that occur during type 2 diabetes and enter directly into the later stage of beta cell failure and hypoinsulinemia. A low dose of STZ alone is usually accompanied by a high rate of induction failure, whereas a high dose causes severe diabetes and a high mortality rate [1], [3],[5]

Comment 3: What are the reasons for choosing the STZ dose of 35 mg/kg?"

Response 3: Srinivasan et al. have previously documented comparable findings, demonstrating that the administration of STZ at varying doses (25 mg/kg, 35 mg/kg, 45 mg/kg, and 55 mg/kg) in a single injection can be employed to establish a diabetic model. Given that the administration of STZ at a dosage of 25 mg/kg did not result in notable hyperglycemia, it was observed that fat-fed/STZ diabetic rats (administered with 45 and 55 mg/kg, IP) displayed considerably elevated glucose levels and a substantial decrease in body weight. Consequently, a dosage of 35 mg/kg for STZ injection was ultimately determined as the optimal dose. However, it is worth noting that the success rate of this approach has not been documented [1]. Also as per Okoduwa et al. a combination of margarine and fructose fortified diet-fed with single low dose (≤35 mg/kg bw.) STZ-treated rat is a suitable non-genetic model for T2D studies. The model mimics the natural history and metabolic features of human T2D. In addition, it is cheap and easy to develop [6].

Comment 4: Is a single administration of 35 mg/kg necessary to make an animal diabetic?"

Response 4: A report in male and female mice indicates that injecting a single dose of STZ in non-fasted animals can successfully induce diabetes. There are also reports of injecting STZ in non-fasting male Wistar and Sprague-Dawley rats with successful induction of hyperglycemia [7]. High fat diet can induce T2DM but it takes long time[8].

Comment 5: In sections 3.2 and 3.1, the authors refer to the results in Table 3. The work lacks Table 3 as a result.

Response 5: thanks a lot for your comment , we re-added table 3 for your kind review.

References

  1. Srinivasan K, Viswanad B, Asrat L, Kaul CL, Ramarao P. Combination of high-fat diet-fed and low-dose streptozotocin-treated rat: A model for type 2 diabetes and pharmacological screening. Pharmacological Research. 2005;52[4]:313–20.
  2. Islam MS, Choi H. Nongenetic model of type 2 diabetes: a comparative study. Pharmacology. 2007;79[4]:243–9.
  3. Skovsø S. Modeling type 2 diabetes in rats using high fat diet and streptozotocin. Journal of Diabetes Investigation. 2014;5[4]:349–58.
  4. Barrière DA, Noll C, Roussy G, Lizotte F, Kessai A, Kirby K, et al. Combination of high-fat/high-fructose diet and low-dose streptozotocin to model long-term type-2 diabetes complications. Scientific reports. 2018 Jan;8[1]:424.
  5. Furman BL. Streptozotocin-Induced Diabetic Models in Mice and Rats. Current protocols in pharmacology. 2015 Sep;70:5.47.1-5.47.20.
  6. Okoduwa SIR, Umar IA, James DB, Inuwa HM. Appropriate insulin level in selecting fortified diet-fed, streptozotocin-treated rat model of type 2 diabetes for anti-diabetic studies. PLoS ONE. 2017;12[1]:1–21.
  7. Ghasemi A, Jeddi S. Streptozotocin as a tool for induction of rat models of diabetes: a practical guide. EXCLI journal. 2023;22:274–94.
  8. Zhao Y, Wang Q-Y, Zeng L-T, Wang J-J, Liu Z, Fan G-Q, et al. Long-Term High-Fat High-Fructose Diet Induces Type 2 Diabetes in Rats through Oxidative Stress. Nutrients [Internet]. 2022;14[11]. Available from: https://www.mdpi.com/2072-6643/14/11/2181

Reviewer 2 Report

Comments and Suggestions for Authors

The study was prepared comparatively well, while several parts could be corrected carefully to improve the paper.

1.      Both drugs affect the inflammation. Could you have the data of inflammation? If so, the paper would be of interest.

2.      A combination therapy can induce some increase of adverse effects and problematic polypharmacy in clinical settings. It may be discussed.

3.      Abstract; the expression ‘monotherapy and combined on’ (Background) could be changed to ‘monotherapy and combination therapy on’ because ‘combination therapy’ was often used in the other parts.

4.      Abstract; the expression ‘lipid profile’ (Methods) could be changed to ‘lipid profiles’.

5.      Abstract; the expression ‘a substantial difference’ (Results) could be detailed more concretely.

6.      Abstract; ‘a’ could be deleted in the expression ‘a combination therapy’ (Results) following the expression of the other parts.

7.      Abstract; the expression ‘genetics expression’ (Conclusion) could be changed to ‘genetic expression’ or ‘genetic and epigenetic expression’(following the expression within the text).

8.      Introduction; after [5] (2nd para), the space seemed to be wide. Some parts should be checked throughout the text because the similarly wide space between the sentences was seen in the text (e.g., seen in the Methods and Discussion sections).

9.      Introduction; the first, third and fourth sentence (7nd para) could respectively have some literature.

10.   Introduction; the first sentence (8nd para) could respectively have some literature.

11.   Introduction; the expression ‘type 2 diabetes’ in the last para could be changed to ‘T2DM’.

12.   Methods; the accuracy (assay CV) of glucose, insulin, and lipids used in the study should respectively be detailed.

13.   Statistics; Pearson rank correlation (methods) or pearson correlation (footnote in Table 4)? There was the inconsistent expression.

14.   Table 2; ‘n = ?’(footnote) may be misprinted. And, ‘n =’ could be changed to ‘n=’(following the expression of other parts).

15.   Table 2; the respective measures might not be accurate to 2 decimal places (reconsider it).

16.   Discussion; the first and second sentence (1st para) could respectively have some literature.

17.   Discussion; the first and second sentence (1st para) could have the expression focused on T2DM (not diabetes mellitus).

18.   Discussion; the first sentence (7nd para) could have some literature.

19.   Discussion; did the first sentence (7nd para) mean that the results were obtained when the respective drugs were used ‘alone’ (or ‘combination’)?

20.   Discussion; In the first sentence (8nd para), ‘in-vitro’ could be changed in italic ‘in vitro’.

21.   Native check should be required in resubmission.

Comments on the Quality of English Language

Native check should be required again.

Author Response

Dear Reviewer 2,

We want to thank you for your insightful comments on our manuscript titled "The Safety and Efficacy of Combining Saxagliptin and Pioglitazone Therapy in Streptozocin-Induced Diabetic Rats" (Manuscript ID: biomedicines-2728012).

We appreciate the time and effort you have dedicated to providing us with feedback on our paper, and we are grateful for the insightful comments and helpful suggestions for improving our manuscript. Considering your questions and requests, we were able to revise and incorporate the necessary amendments into our manuscript. Please find below our point-by-point response to the comments. We have also uploaded the revised manuscript, with amendments highlighted in yellow. Please note that we sometimes talk about lines that aren't highlighted because they are part of the answer.

We look forward to hearing from you soon and responding to any further questions and comments you may have.

Sincerely,

Ahmed Othman

This file contains all comments provided by reviewer two and the authors' responses.

  • Comment 1 : “Both drugs affect the inflammation. Could you have the data of inflammation? If so, the paper would be of interest.”

Response 1: Thank you for your insightful comments. Table three is added to include antioxidant markers. The cellular antioxidant and anti-inflammatory defense mechanism showed a significant decrease in levels of SOD (2031 vs. 4680.14 U/g tissue), catalase (55.16 vs. 121.03 U/g tissue), and GSH (14.02 vs. 37.0 mmol/g tissue), and in liver homogenate a significant increase in MDA (429.89 vs. 259.34 nmol/g tissue) in the STZ-induced diabetic group compared to the normal control group at P Ë‚ 0.001. However, treatment with combination therapy of Saxagliptin and Pioglitazone significantly improves the oxidative stress parameters better than Saxagliptin or Pioglitazone monotherapy. The group treated with Saxagliptin and Pioglitazone showed a significant increase in levels of SOD, catalase, and GSH and a significant decrease in MDA at P Ë‚ 0.001 compared to the groups treated with either Pioglitazone or Saxagliptin (Table 3).

Groups

Urea (mg/dL)

Creatinine (mg/dL)

ALT

(U/L)

AST

(U/L)

SOD

(U/gm tissue)

Catalase (U/g tissue)

MDA

(nmol/g tissue)

GSH

(mmol/g tissue)

Normal control

41.16 ± 4.19

0.73 ± 0.04

21.45 ± 1.38

59.39 ± 2.35

4680.14 ± 307.31

121.03 ± 4.74

259.34 ±11.29

37.0 ± 2.31

Diabetic control

109.61 ±4.71

a, c, d, e

1.52 ± 0.11

a, c, d, e

39.35 ± 1.59

a, c, d, e

94.91 ± 3.19

a, c, d, e

2031 ± 193.02

a, c, d, e

55.16 ± 3.02

a, c, d, e

429.89 ±11.37

a, c, d, e

14.02 ± 1.21

a, c, d, e

Pioglitazone group

91.33 ± 3.74

a, b, d, e

0.9 ± 0.08

b

25.19 ± 1.31

b

71.22 ± 3.42

a, b, e

3502.75 ± 63.59

a, b

86.65 ± 2.14

a, b, d, e

339.12 ± 4.93

a, b, d, e

26.25 ± 0.89

a, b, d, e

Saxagliptin

group

70.03 ± 4.72

a, b, c

0.81 ± 0.05

b

28.84 ± 0.8

a, b, e

74.59 ± 1.046

a, b, e

2972.92 ± 272.97

a, b, e

72.47 ± 2.45

a, b, c, e

382.86 ± 5.84

a, b, c, e

20.16 ± 1.27

a, b, c, e

Saxagliptin +Pioglitazone

61.26 ± 4.12 a, b, c

0.78 ± 0.07

b

22.09 ± 1.61

b, d

60.64 ±. 0.54

b, c, d

4297.49 ±187.71

b, d

101.4 ± 2.31

a, b, c, d

302.2 ± 3.34

a, b, c, d

33.68 ± 1.06

b, c, d

Also, IL1B Expression was added to Figure 2 as per section 3.3.  Lines 5 and 6

The treated groups exhibited a significant upregulation of PIK3R1 expression and downregulation of PCK1, IL1B, and MIR 29a compared to the diabetic control rats. (Figure 2). 

These are all data obtained regarding antioxidants or inflammation.

  • Comment 2: “A combination therapy can induce some increase of adverse effects and problematic polypharmacy in clinical settings. It may be discussed.”

Response 2: In the Introduction section, the 8th paragraph at 5, 6, 7, and 8 Lines were added to discuss previous Literature.

In 76 weeks trial on patients with type 2 diabetes mellitus, there was no rise in the cardiac or cardiovascular safety signals in the groups receiving saxagliptin and thiazolidinediones medication [1]. Also, there is no Drug-Drug Interaction between Pioglitazone and Saxagliptin. They don't require changing the saxagliptin dosage to be taken at the same time [2].

  • Comment 3: Abstract; the expression ‘monotherapy and combined on’ (Background) could be changed to ‘monotherapy and combination therapy on’ because ‘combination therapy’ was often used in the other parts.”

Response 3: Thanks a lot for this comment. In the upgraded version of the manuscript, we tried to make consistency throughout the manuscript, and we addressed the comment.

 This study aimed to assess the effect of Saxagliptin and Pioglitazone monotherapy and combination therapy on the biochemical and biological parameters in streptozotocin (STZ) induced diabetic rats.

  • Comment 4: Abstract; the expression ‘lipid profile’ (Methods) could be changed to ‘lipid profiles.

Response 4: Thanks a lot for this comment. In the upgraded version of the manuscript, we tried to make consistency throughout the manuscript, and we addressed the comment.

For a one-month duration, rats were divided into five groups. Glucose homeostasis traits, lipid profiles, kidney functions, liver enzymes, and oxidative stress markers were measured.

  • Comment 5: Abstract; the expression ‘a substantial difference’ (Results) could be detailed more concretely.

Response 5: Thank you for your point. I agree with your opinion. For that reason, we replace it with the following: " Combination Therapy improves oxidative stress markers more than either drug alone. The combination therapy had significantly higher levels of SOD, catalase, and GSH and lower levels of MDA compared to the monotherapy."

  • Comment 6: Abstract; the expression ‘a substantial difference’ (Results) could be detailed more concretely.

Response 6: Thank you for this comment. The comment was applied, and an extensive language check was done.

However, combination therapy of Saxagliptin and Pioglitazone was more effective than either Saxagliptin or Pioglitazone alone in reversing these results, especially for PEPCK and IL-1 β.

  • Comment 7: Abstract; the expression ‘genetics expression’ (Conclusion) could be changed to ‘genetic expression’ or ‘genetic and epigenetic expression’(following the expression within the text).

Response 7: Thank you for pointing this out. The comment was addressed, and in the upgraded version of the manuscript, we tried to make it consistent throughout. You can find it highlighted throughout the Manuscript.

  • Comment 8: Introduction; after [5] (2nd para), the space seemed to be wide. Some parts should be checked throughout the text because the similarly wide space between the sentences was seen in the text (e.g., seen in the Methods and Discussion sections).

Response 8: thanks a lot for your notification; an extensive review regarding formatting was done, and we addressed this comment.

  • Comment 9: Introduction: the first, third and fourth sentence (7nd para) could respectively have some literature.

Response 9: Thank you for your point. We added 3 additional literatures to the previous sentences as follows.

Saxagliptin is a selective dipeptidyl peptidase-4 (DPP-4) inhibitor [3]. It increases glucagon-like peptide-1 levels, motivates insulin production, and lowers glucose levels by activating the incretin hormones, glucagon-like peptide-1, and glucose-dependent insulinotropic polypeptide [4]. Pioglitazone, an insulin sensitizer, is one of the thiazolidinediones (TZD), and it is a peroxisome proliferator-activated receptor-gamma (PPAR-γ) agonist [5]. Pioglitazone enhances insulin sensitivity as it regulates the expression of many genes concerned with lipid and carbohydrate metabolism [6].

  • Comment 10: Introduction; the first sentence (8nd para) could respectively have some literature. Response 10: Thank you for your point. We added additional literature to the first sentence as follows.

Previous literature showed that adding DPP-4 inhibitors to Pioglitazone was well tolerated, did not cause hypoglycemia, and did not significantly worsen Pioglitazone-induced weight gain [7].

  • Comment 11: Introduction; the expression ‘type 2 diabetes’ in the last para could be changed to ‘T2DM’.

Response 11: Thank you for your point. The expression has been changed to T2DM In the mentioned part and throughout the Manuscript.

  • Comment 12: Methods: the accuracy (assay CV) of glucose, insulin, and lipids used in the study should respectively be detailed.

Response 12: Thank you very much for your point. Actually two replicates were measured per each sample for all tests done in the study, a more detailed procedure attached to the manuscript

Serum glucose level was measured by the enzymatic colorimetric method using (Biodiagnostic kit, Egypt, Catalog No. GLU109240) was used, performed in two replicates Gluocose concentration was obtained by the following equation, Glucose concentration= (Asample / Astandard) x Standard Conc. Serum insulin was determined using a rat insulin (INS) ELISA kit (Wuhan Fine Biotech Co. Ltd., China, Catalog No. ER1113) was used, performed in two replicates (standard curve range: 78.125-5000pg/ml, sensitivity: < 46.875 pg/ml, intra-assay: CV < 8%, inter-assay: CV < 10%) as per manufacturer procedures. Two indirect indices were calculated. First, the Homeostasis Model Assessment-Insulin Resistance (HOMA-IR) was calculated using the equation: (glucose x insulin) / 405 [8]. Second, the quantitative insulin sensitivity check index (QUICKI) was calculated using the equation 1/ (log insulin + log glucose) [9].

Lipid profiles, including serum triglycerides (TG), total cholesterol (TC), and high-density lipoprotein cholesterol (HDL-C), were measured in two replicates by enzymatic colorimetric methods using [Biodiagnostic kits, Egypt, (Catalog No. TG117249), (Catalog No.CH0104200), and (Catalog No. MG266001) respectively]. Low-density lipoprotein cholesterol (LDL-C) was calculated according to Friedewald's

  • Comment 13: Statistics; Pearson rank correlation (methods) or pearson correlation (footnote in Table 4)? There was an inconsistent expression.

Response 13: Thank you for your point. It's changed to Pearson rank correlation in the mentioned part and throughout the Manuscript.

  • Comment 14: Table 2; ‘n =?’(footnote) may be misprinted. And, ‘n =’ could be changed to ‘n=’(following the expression of other parts).

Response 14: thanks a lot for your notification; an extensive review regarding formatting was done, and we addressed this comment. As follows: n=7 rats

  • Comment 15: Table 2; the respective measures might not be accurate to 2 decimal places (reconsider it).

Response 15: Thank you very much regarding this comment which make us go deep and deep in the ways of reporting and measures, it is really important comment, we found the following as per Straseski et al., the overall distribution of the data was unchanged with the use of either 2 or 3 decimal places[10]. Also, as per Sinnott et al., Results suggest that as the number of decimal places exceeds two, the number of false results increases [11], [16].

  • Comment 16: Discussion; the first and second sentence (1st para) could respectively have some literature.

Response 16: Thank you for your point. We added additional literature to previously mentioned sentences as follows.

T2DM requires effective therapeutic targets [12]. The proper management of T2DM may be well achieved using combination therapy rather than monotherapy [13].

  • Comment 17: Discussion; the first and second sentences (1st para) could have the expression focused on T2DM (not diabetes mellitus).

Response 17: Thank you for your point. The expression has been changed to T2DM In the mentioned part and throughout the Manuscript.

  • Comment 18: Discussion; the first sentence (7nd para) could have some literature.

Response 18: Thank you for your point. We added additional literature to previously mentioned sentence as follows.

Pioglitazone and Saxagliptin Severally have previously been shown to reduce inflammation and alter inflammatory cytokine (IL-1β and IL-6) levels [14], [15].

  • Comment 19: Discussion; did the first sentence (7nd para) mean that the results were obtained when the respective drugs were used ‘alone’ (or ‘combination’)?

Response 19: Thank you for your point. This sentence indicates when the respective drugs were used ‘alone.’ 

Pioglitazone and Saxagliptin Severally have previously been shown to reduce inflammation and alter inflammatory cytokine (IL-1β and IL-6) levels.

We already added additional litratures

Pioglitazone and Saxagliptin Severally have previously been shown to reduce inflammation and alter inflammatory cytokine (IL-1β and IL-6) levels [14], [15].

  • Comment 20: Discussion; In the first sentence (8nd para), ‘in-vitro’ could be changed in italic ‘in vitro’.

Response 20: Thank you for your point. The expression has been changed to italic ‘in vitro’.

In the mentioned part and throughout the Manuscript.

  • Comment 21: Native check should be required in resubmission.

Response 21: Thank you for your recommendation. We have performed comprehensive language editing, checking for grammatical errors and typos. Some sentences have been paraphrased to make them more understandable.

References

  1. Hollander PL, Jia Li, Frederich R, Allen E, Chen R. Safety and efficacy of saxagliptin added to thiazolidinedione over 76 weeks in patients with type 2 diabetes mellitus. Diabetes and Vascular Disease Research [Internet]. 2011;8[2]:125–35. Available from: http://journals.sagepub.com/doi/10.1177/1479164111404575
  2. Patel CG, Kornhauser D, Vachharajani N, Komoroski B, Brenner E, Handschuh del Corral M, et al. Saxagliptin, a potent, selective inhibitor of DPP-4, does not alter the pharmacokinetics of three oral antidiabetic drugs (metformin, glyburide or pioglitazone) in healthy subjects. Diabetes, Obesity and Metabolism. 2011;13[7]:604–14.
  3. Shubrook J, Colucci R, Guo A, Schwartz F. Saxagliptin: A Selective DPP-4 Inhibitor for the Treatment of Type 2 Diabetes Mellitus. Clinical medicine insights Endocrinology and diabetes. 2011;4:1–12.
  4. Dave DJ. Saxagliptin: A dipeptidyl peptidase-4 inhibitor in the treatment of type 2 diabetes mellitus. Journal of pharmacology & pharmacotherapeutics. 2011 Oct;2[4]:230–5.
  5. Orasanu G, Ziouzenkova O, Devchand PR, Nehra V, Hamdy O, Horton ES, et al. The peroxisome proliferator-activated receptor-gamma agonist pioglitazone represses inflammation in a peroxisome proliferator-activated receptor-alpha-dependent manner in vitro and in vivo in mice. Journal of the American College of Cardiology. 2008 Sep;52[10]:869–81.
  6. Coletta DK, Sriwijitkamol A, Wajcberg E, Tantiwong P, Li M, Prentki M, et al. Pioglitazone stimulates AMP-activated protein kinase signalling and increases the expression of genes involved in adiponectin signalling, mitochondrial function and fat oxidation in human skeletal muscle in vivo: a randomised trial. Diabetologia. 2009 Apr;52[4]:723–32.
  7. Mikhail N. Combination therapy with DPP-4 inhibitors and pioglitazone in type 2 diabetes: Theoretical consideration and therapeutic potential. Vascular Health and Risk Management. 2008;4[6]:1221–7.
  8. Matthews DR, Hosker JP, Rudenski AS, Naylor BA, Treacher DF, Turner RC. Homeostasis model assessment: insulin resistance and beta-cell function from fasting plasma glucose and insulin concentrations in man. Diabetologia. 1985 Jul;28[7]:412–9.
  9. Gutch M, Kumar S, Razi SM, Gupta KK, Gupta A. Assessment of insulin sensitivity/resistance. Indian journal of endocrinology and metabolism. 2015;19[1]:160–4.
  10. Straseski JA, Whale C, Wilson A, Strathmann FG. The significance of reporting to the thousandths place: Figuring out the laboratory limitations. Practical laboratory medicine. 2017 Apr;7:1–5.
  11. Sinnott M, Eley R, Steinle V, Boyde M, Trenning L, Dimeski G. Decimal numbers and safe interpretation of clinical pathology results. Journal of clinical pathology. 2014 Feb;67[2]:179–81.
  12. Su J, Luo Y, Hu S, Tang L, Ouyang S. Advances in Research on Type 2 Diabetes Mellitus Targets and Therapeutic Agents. International journal of molecular sciences. 2023 Aug;24[17].
  13. Gudoor R, Suits A, Shubrook JH. Perfecting the Puzzle of Pathophysiology: Exploring Combination Therapy in the Treatment of Type 2 Diabetes. Diabetology [Internet]. 2023;4[3]:379–92. Available from: https://www.mdpi.com/2673-4540/4/3/32
  14. Kaplan J, Nowell M, Chima R, Zingarelli B. Pioglitazone reduces inflammation through inhibition of NF-κB in polymicrobial sepsis. Innate immunity. 2014 Jul;20[5]:519–28.
  15. Zeng X, Li X, Chen Z, Yao Q. DPP-4 inhibitor saxagliptin ameliorates oxygen deprivation/reoxygenation-induced brain endothelial injury. American journal of translational research. 2019;11[10]:6316–25.
  16. Burtis, C.A. and Bruns, D.E. (2015) Tietz Fundamentals of Clinical Chemistry and Molecular Diagnostics. 7th Edition, Elsevier Missouri, Saunders, 364-370.

Round 2

Reviewer 1 Report

Comments and Suggestions for Authors

The authors responded to all the comments raised and the quality of the work improved considerably.

I consider the article suitable for publication in Biomedicines. 

Reviewer 2 Report

Comments and Suggestions for Authors

The paper has been much improved.